# The structure of a red-shifted photosystem I reveals a red site in the core antenna

Hila Toporik [1,2], Anton Khmelnitskiy[3], Zachary Dobson [1,2], Reece Riddle [1,2], Dewight Williams[4], Su Lin[1,2,5], Ryszard Jankowiak[3,6] & Yuval Mazor [1,2 ✉]

Photosystem I coordinates more than 90 chlorophylls in its core antenna while achieving near perfect quantum efficiency. Low energy chlorophylls (also known as red chlorophylls) residing in the antenna are important for energy transfer dynamics and yield, however, their precise location remained elusive. Here, we construct a chimeric Photosystem I complex in *Synechocystis* PCC 6803 that shows enhanced absorption in the red spectral region. We combine Cryo-EM and spectroscopy to determine the structure−function relationship in this red-shifted Photosystem I complex. Determining the structure of this complex reveals the precise architecture of the low energy site as well as large scale structural heterogeneity which is probably universal to all trimeric Photosystem I complexes. Identifying the structural elements that constitute red sites can expand the absorption spectrum of oxygenic photosynthetic and potentially modulate light harvesting efficiency.

[1] School of Molecular Sciences, Arizona State University, Tempe, AZ 85287-1604, USA. [2] Biodesign Center for Applied Structural Discovery, Arizona State University, Tempe, AZ 85287, USA. [3] Department of Chemistry, Kansas State University, Manhattan, KS 66506, USA. [4] John M. Cowley Center for High Resolution Electron Microscopy, Arizona State University, Tempe, AZ 85287, USA. [5] Center for Innovations in Medicine at the Biodesign Institute, Arizona State University, Tempe, AZ 85287, USA. [6] Department of Physics, Kansas State University, Manhattan, KS 66506, USA. ✉email: yuval.mazor@asu.edu

Oxygenic photosynthesis, taking place in cyanobacteria, algae, and plants, powers the biosphere by converting light photons into chemical energy. Photochemical conversion occurs in two photosystems, photosystem I and photosystem II (PSI and PSII respectively), which are multi-subunit protein-pigment complexes located in thylakoid membranes.

The cyanobacterial PSI is composed of a highly conserved core containing 11 subunits that coordinate 94–96 chlorophylls (Chls) and 22 carotenoids (Cars) per PSI monomer. These PSI monomers assemble into trimers or tetramers depending on the species[1–3]. Six of the PSI Chls, located at the center of the complex, make up the internal electron transport chain. Two of these six Chls form a special pair called P700 which is oxidized to the relatively stable P700$^+$ state (with a characteristic bleach around 700 nm) as part of the photochemically induced charge separation process in PSI. The rest of the Chls function as a core antenna for excitation energy transfer (EET). Altogether, PSI functions with near perfect quantum efficiency; ~98% of the photons absorbed in the antenna result in photochemical charge separation in the reaction center[4].

One of the conserved features of PSI is the existence of a small number of "red" or low-energy Chls, defined as Chls with an absorption maximum at longer wavelengths than P700[5–9]. The presence of red sites necessitates uphill energy transfer steps as part of the function of PSI and this is why this group of Chls plays a major part in determining the rate of EET in the PSI antenna[6,10–12]. Specifically, the trapping rate of the cyanobacterial PSI increases in rough proportion to its red sites content, from ~20 ps in complexes nearly devoid of red Chls to ~50 ps in PSI with an extremely red-shifted Chls[10,13]. The large number of pigments in the core antenna and their similar electronic transitions makes the identification of these Chls extremely challenging[12,14].

In solution, Chls can form dimers and higher order aggregates that result in a strong red spectral shift due to excitonic coupling[15]. It was suggested that Chl aggregates, dimers and trimers in the PSI antenna, are the basic structural unit forming red sites in PSI antenna[12,16,17].

Several, not mutually exclusive functions were put forth with regards to the red Chls: It was suggested that red Chls can focus the energy to P700 by reducing the number of EET steps between the bulk Chls and P700[18]. In addition, red Chls comprise up to 10% of the antenna Chls and can contribute to light harvesting by absorbing light in the far-red region, a feature that is especially important in high-density cultures or shaded environments where light is not only scarce but tend to be red shifted due to absorption by multiple cells[19,20]. Another suggestion is that red Chls play a role in photoprotection. It was shown that when P700 is oxidized and the RC is closed, emission from the red sites is significantly reduced, potentially minimizing the formation of Chls triplets in PSI[21,22].

In plants, most of the red sites are located in the peripheral antenna LHCI[8]. Cyanobacteria, which lack the LHCI antenna, carry their red sites in the core antenna of PSI. The properties and numbers of red Chls vary across different species: estimations of the number of red Chls range from five in Synechocystis PCC 6803 (Synechocystis) to ten Chls in Thermosynechococcus elongatus (T. elongatus)[9].

A major breakthrough in identifying the location of red Chls was the elucidation of the PSI structure from T. elongatus[1] which enabled the calculation of dipole interactions among the PSI Chls. This resulted in the first assignment of red Chls in PSI internal antenna[1]. Since then, multiple attempts to model the steady state properties and energy transfer processes in PSI using various treatments were presented[23–30]. However, very little structure–function analysis has been carried out on the core antenna and

this is critical for every modeling attempt. Structures of PSI with known differences in their red sites content are essential in order to reach the correct assignment of the red states in PSI[2].

From the Chls suspected as contributing to the red states in T. elongatus, one unusual Chl trimer stands out as unique in PSI. This Chl trimer is made up of Chls B31-B32-B33 (Chl numbering follows[1]). The assignment of the B31-B32-B33 trimer as a red site is controversial and was disputed by some theoretical studies[26,30]. Different approaches attempted to fit several types of spectroscopic data with site energies values assigned to the different PSI Chls. So far, the calculated site energies differ quite substantially between publications and in particular do not agree on the identity of red pigments in PSI and their site energies[23,25,27,28,30] (In fact, in most cases multiple sets of site energies can be used to fit the data). Overall, more than 30 different Chls were implicated as red in various calculations with a consensus of only a single site (based on data from six different publications; Fig. 1a).

Here we produced a chimeric PSI in cyanobacteria, introducing a small loop which adds a single Chl (B33) to the PSI complex of Synechocystis, reconstituting the B31-B32-B33 Chl trimer from T. elongatus in the Synechocystis PSI. We demonstrate that this additional Chl forms a new red state in Synechocystis PSI (named C710), contributed to by the B31-B32-B33 trimer. We measure the effect of modifying a red state in PSI on the growth of cells and the energy transfer processes in PSI. We solved the structure of the chimeric complex using single particle Cryo-EM and measured a new type of large-scale heterogeneity in the PSI trimer, suggesting that each individual PSI monomer can occupy a range of positions in the PSI trimer. The location of low-energy sites in the core antenna and the large-scale heterogeneity discovered in this study are universal properties of the cyanobacterial PSI.

## Results

**Incorporation of a new low-energy chlorophyll into the Synechocystis PSI complex.** The structure of PSI from T. elongatus contains 96 Chl molecules coordinated by 12 protein subunits. All the different Chls suggested to contribute to the low-energy states in this system are colored in red in Fig. 1a. Fig. 1b shows an overlay of the Chl networks from T. elongatus and Synechocystis. T. elongatus was shown to contain a larger number of red Chls at lower energies[9]. Most Chl positions are absolutely conserved between T. elongatus and Synechocystis. This similarity extends not only to the position and orientation of the Chl rings, but also to their environments. One notable difference is an additional Chl in the T. elongatus structure, B33. Examination of the protein environment in T. elongatus PSI showed an additional loop coordinating this Chl (Fig. 1c). Sequence alignment showed that this loop is located between conserved amino acids sequences. The extra loop sequence from T. elongatus was inserted into the Synechocystis PsaB gene to form the Red_a PSI complex (Fig. 1d).

**Addition of the B33 Chl to PSI increases red emissions in cells.** Initial indications that Chl B33 contributes to a red state in PSI were observed in cells. The fluorescence spectra of Red_a cells excited at the Chl Soret band (440 nm) showed a shoulder in the red region of the spectrum in comparison to the wild type (WT; Fig. 2a). Examining the emission at 77 K showed that the main red emission peak, which originates from the PSI red states, is red shifted from 722 nm to 724 nm. Additional peaks in the 77-K emission spectra, originating from PSII (684 nm and 694 nm) and phycobilisomes (648 nm), did not change their position, showing that only PSI was affected by the Red_a mutation (Fig. 2b). We compared the growth of cells carrying the Red_a mutation at

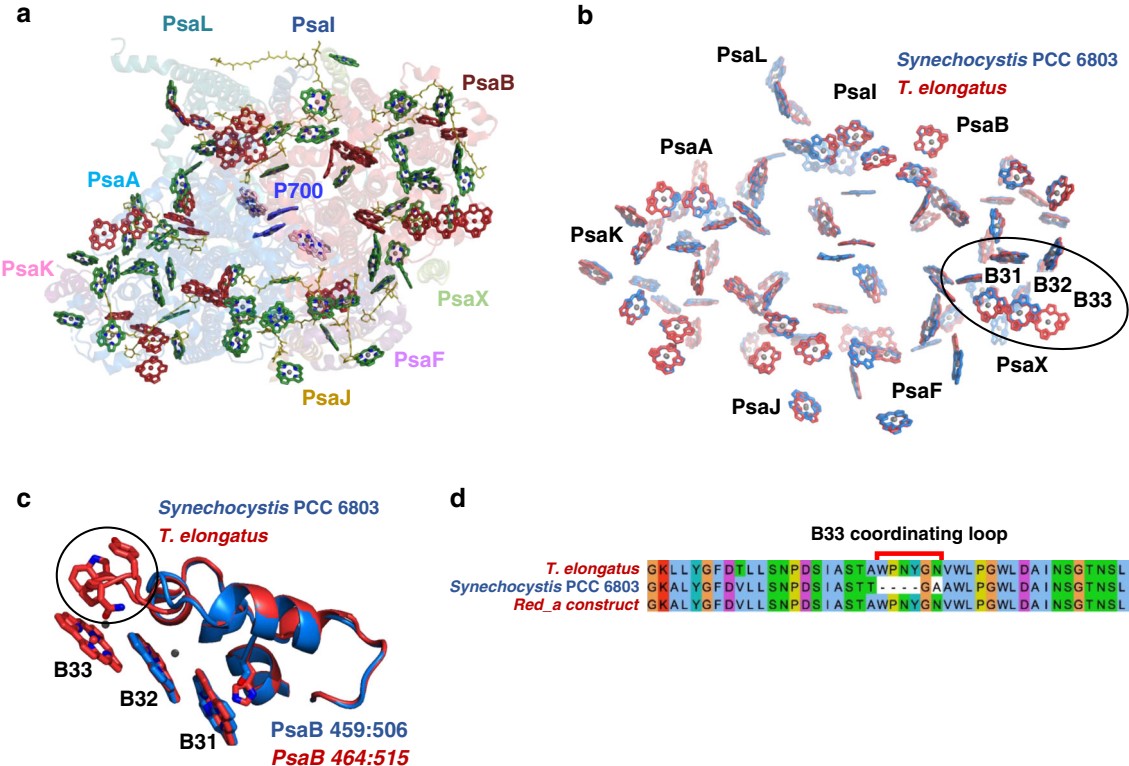

**Fig. 1 Structural differences in chlorophyll positions between *T. elongatus* and *Synechocystis* PSI complexes. a** The structure of a *T. elongatus* monomer PSI complex - core antenna Chls are shown in green, previous suggestions for red Chls are in red, β-carotenes in olive, and P700 Chls and the electron transport chain accessory Chls are in blue and pink, respectively. **b** Superposition of the PSI core Chls from *T. elongatus* (in red) on *Synechocystis* (in blue), the close similarity in positions results in mixed red/blue coloring on most Chls. Black ellipse marks the putative red site, B31-B32-B33 Chl trimer. **c** A close up of the red loop structure from both cyanobacterial species shows the highly conserved structure outside of a short loop coordinating Chl B33. **d** Alignments of PsaB sequences from *Synechocystis*, *T. elongatus*, and the Red_a mutant.

various light intensities to WT cells and strains carrying only the resistance gene and observed no difference in the growth of cells over a wide range of light intensities showing that the Red_a PSI is functional in cells (Fig. 2c).

The Red_a mutation did not affect the oligomeric state of the purified PSI complex (Fig. 3a) as we observed the same trimer to monomer ratio in solubilized membranes. In addition, the same subunit composition is observed in both Red_a and WT PSI, we conclude that the Red_a mutation does not interfere with the native structure of the complex (Fig. 3b). As seen in Fig. 3c, the absorption spectra of PSI complexes from WT and Red_a strains overlap closely in the visible spectral region. The difference spectrum shows a major positive peak in the $Q_y$ region of PSI, centered ~709.3 nm. This clearly indicates that the added B33 Chl contributes to one of the lowest-energy states in PSI and absorbs light at longer wavelengths than 700 nm (Fig. 3d). Gaussian approximation of the difference spectra was maximal at ~709.3 nm and shows a large full width at half maximum (fwhm) of ~25 nm (450 cm$^{-1}$), this together with the intensity of the band suggests that more than one Chl molecule is involved, in agreement with previous experiments suggesting that Chl aggregates constitute the red states in PSI. In the 350–520-nm region, a negative signal with a clear Car signature is observed. We attribute this signal to a loosely bound carotene population present in the WT trimer. To test this, we subjected our samples to a further step of gentle purification, a repeat of the sucrose gradient purification. This test showed that the difference in the carotene region is easily lost while the difference in the chlorophyll $Q_y$ region is nearly identical after this extra purification step (Supplementary fig. 1c, d). The fluorescence

spectra of the purified complexes at room temperature clearly show an increase in the Red_a complex emission in the red part of the spectrum compared to WT (Fig. 3e), the differences between the WT and Red_a peaks near 720 nm. At 77 K, emission from the red states is clearly observed with a fluorescence band maximum of 722 nm and 724 nm observed in the WT and Red_a, respectively (Fig. 3f). To get a more precise view of the differences between WT and Red_a, we measured the emission from both complexes at 4 K (Supplementary fig. 2a). The results show that the emission contributed by the B31-B32-B33 Chl trimer (assigned below to the C710 state) peaks near 725 nm (at 4 K) and is extremely broad (fwhm ~460 cm$^{-1}$), suggesting a larger degree of mixing with the charge-transfer (CT) state(s) than that observed for the B31-B32 dimer present in WT PSI.

**Low temperature studies show that B31-B32-B33 contributes to the C710 state in PSI.** To minimize thermal broadening and get a clearer picture of the changes introduced into the Red_a mutant we carried out low-temperature absorption and non-resonant hole burning (HB) measurements. Figure 4 shows the 4-K absorption spectra for the Red_a (red spectrum, curve **a**) and WT PSI (blue spectrum, curve **b**). Both spectra were normalized in the $Q_y$ region to account for their Chl content.

Similarly, to the observed differences in the room temperature absorption spectra, the absorption difference between the Red_a mutant and WT PSI (within the lowest-energy band) accounts for an additional absorption of about two Chls per PSI monomer, not one Chl as expected in Red_a. The difference between 4 K absorption spectra **a** and **b** revealed a positive band with a maximum near 710.4 nm with fwhm of ~13 nm (260 cm$^{-1}$).

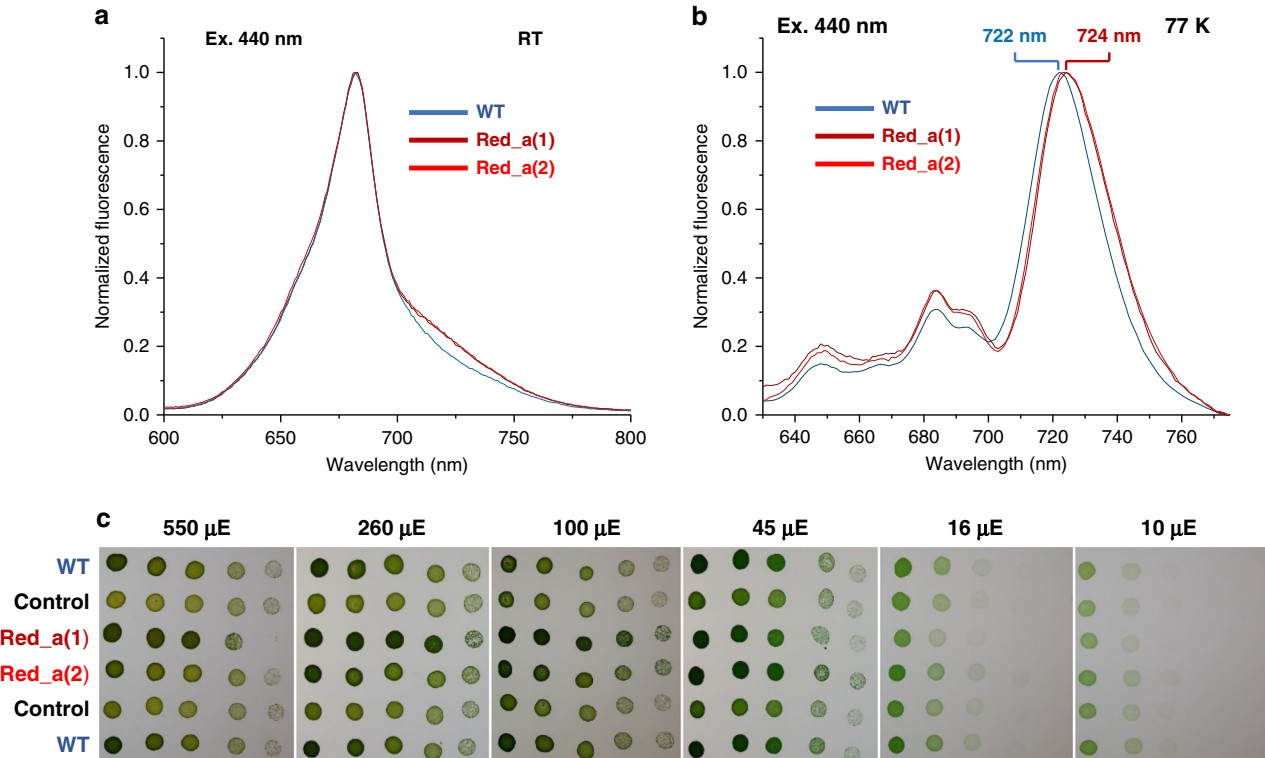

**Fig. 2 Spectroscopic properties and physiological effect of Red_a in cells. a, b** Fluorescence emission from WT (blue) and two independent Red_a isolates (Red_a(1/2)) cells at room temperature (**a**) and 77 K (**b**). **c** No growth differences between WT and Red_a cells were found over a range of light intensities. (5-fold serial dilutions of each strain were incubated on BG-11 plates without added glucose, pictures were taken after 10 days of incubation at 30 °C). Control lines contain the antibiotic resistance gene used in the Red_a strains but do not carry the Red_a mutation.

Based on the Cryo-EM structure of the Red_a mutant (see below), we can exclude that more than one Chl was incorporated per monomer. Therefore, we suggest that the oscillator strength of the low-energy states has increased due to intensity stealing. Since the B31-B32 dimer in *Synechocystis* was previously assigned to the C706 state[31], we confirm that this state is indeed one of the low-energy states in WT PSI. Due to addition of B33 Chl to the B31-B32 dimer, and increased interaction between the three Chls, the maximum of this state shifts to 710.4 nm. This new absorption band is assigned to the C710 state in the reminder of the manuscript. That is, this state is contributed to by the B31-B32-B33 Chl trimer. Previously, HB measurements revealed the lowest-energy state in this system is actually located at 714 nm (termed C714 trap)[31]. The nonresonant HB spectrum of the WT PSI complex (Fig. 4, curve c), shows the typical wide hole burned in this complex, centered at 714 nm, which mostly represents the C714 state[31]. In contrast, in the Red_a PSI complex the nonresonant HB spectrum (Fig. 4, curve d) shows very broad hole with the minimum near 710 nm (indicated by the asterisk in Fig. 4) and assigned to the C710 state. The broad hole includes contribution from the C714 state as well[32], indicating that in the Red_a PSI complex, excitation energy is not efficiently transferred to the lowest-energy C714 state at low temperatures. We anticipate that future frequency-dependent resonant hole-burning spectra obtained for several PSI mutants will shed even more light on the EET dynamics in WT and mutated PSI complexes. Assignment of low-energy states (and their composition) in WT PSI and its Red_a mutant is summarized in Table 1.

**Trapping is affected in the Red_a mutant at room temperature.** To monitor EET in PSI we measured the fluorescence decay

kinetics of purified WT and Red_a mutant PSI trimers in the wavelength region of 630–800 nm at room temperature using a streak camera setup. The overall emissions summed over the duration of the measured time window follows the general shape of the steady state emission spectra, with Red_a showing enhanced emission in the red region of the emission profile (Fig. 5a). The fluorescence decay was fitted with three decay components using global analysis[33,34]. Total decay summed over the 630–800-nm range is shown in Fig. 5b (complete data, fit, and residuals can be found in Supplementary fig. 3) and clearly show slower decay in the Red_a PSI complex. Decay associated spectra (DAS) obtained from global analysis show a typical fast component with decay in the blue region of the spectrum (positive decay amplitudes) and a rise component in the red region (negative decay amplitudes; Fig. 5c), originating from EET from the bulk antenna Chls to the red Chls as the core antenna equilibrates.

The second decay component is mostly associated with trapping processes and also includes contributions from bulk antenna Chls and the different red Chl pools (Fig. 5d). In addition, a low amplitude, slow component associated with free pigments (Fig. 5e) is present in the data. While the fitted lifetimes of the fast components appear very different (4 and 8 ps in the WT and Red_a complex, respectively) the differences should not be considered as significant as they are comparable to half of the instrument response function (IRF) of our setup (Fig. 5b, c). In addition, Gaussian decomposition of the transfer components showed that the transfer processes from the major antenna to the red pools in WT and Red_a complexes are similar (Fig. 5c). In contrast, the trapping component in Red_a is significantly slower than the WT trapping time (35 ps vs 28 ps). In addition, Gaussian decomposition of the DAS showed that the main differences between WT and Red_a PSI complexes lay in the amplitudes of

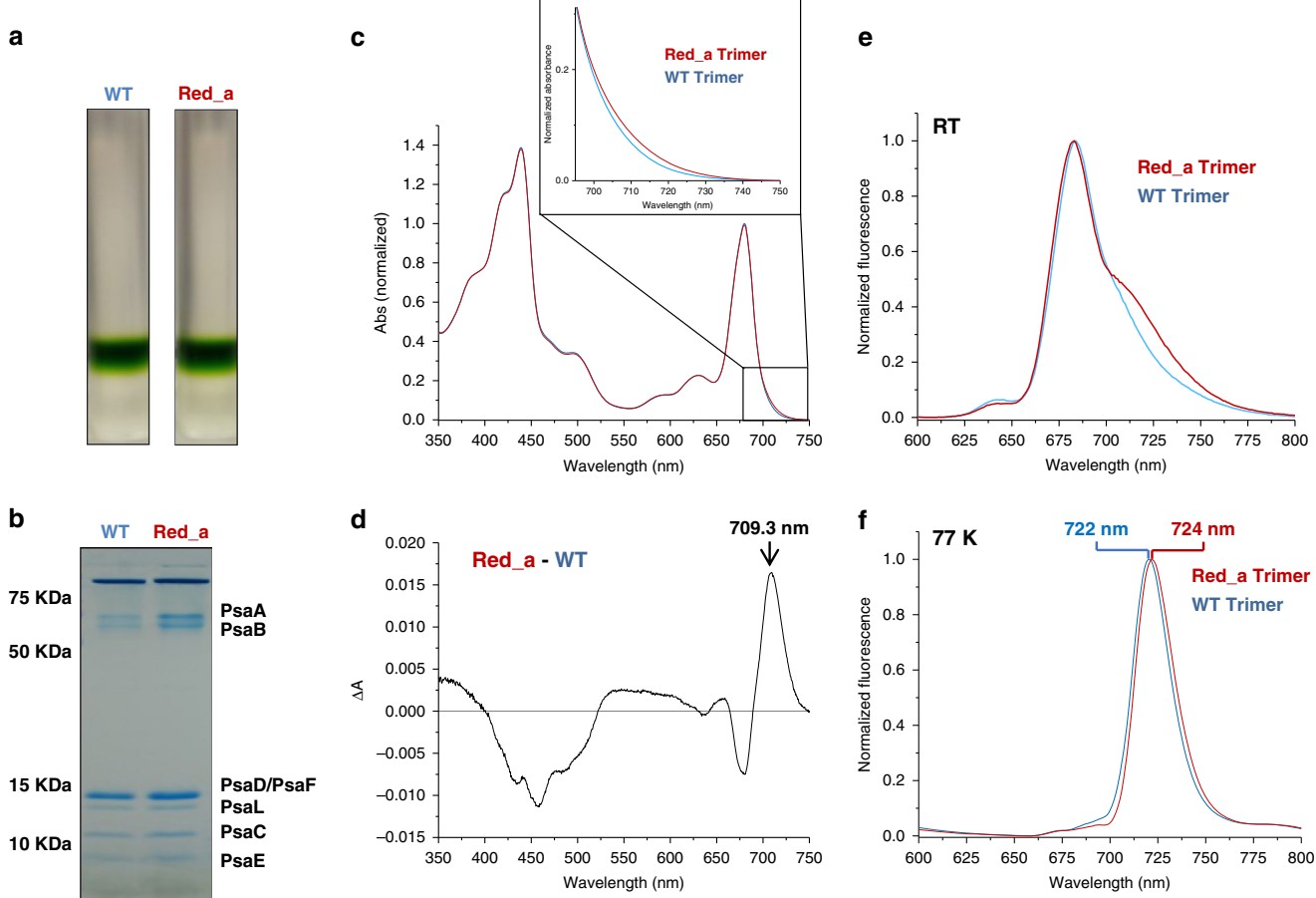

**Fig. 3 Adding a red chlorophyll site to PSI. a** The WT and Red_a isolated complexes were loaded on 10–30% sucrose gradient **b** SDS-PAGE of the isolated WT and Red_a PSI complexes from the sucrose gradient shown in **a**. **c** Room temperature absorption spectra of WT (blue line) and Red_a (red line) PSI trimers. The insert shows the enlarged 695–750 nm spectral region. **d** Difference absorption spectrum (Red_a – WT) shows a clear band centered at 709.3 nm. **e** Room temperature emission spectra of WT and Red_a PSI trimers showing the enhanced red emission from the Red_a PSI complex (705–760 nm). **f** Emission spectra taken at 77 K showing a red shift in the emission maximum in the Red_a PSI (maximum at 724 nm) compared to the WT PSI (maximum at 722 nm).

the 685–687-nm component which decreases significantly in Red_a. Accompanied with the blue band decrease, an increase of the 707–709 nm component and a slight increase of the 717-nm component was observed, suggesting an excitation redistribution which is more towards the newly formed red states (B31-B32-B33) in Red_a. The location of the B31-B32-B33 trimer within the Red_a PSI complex implies that trapping directly from the B31-B32-B33 trimer is unlikely. The Red_a "trapping" DAS must include an uphill energy transfer to the bulk antenna at room temperature. Such process is extremely fast and is manifested via the decrease in the decay amplitude in the blue region of the trapping DAS together with the rise in decay amplitudes in the red region of the trapping DAS. Altogether, large effect on trapping time is observed by the addition of a single Chl to the PSI antenna. C710 location, at the edge of the core antenna, where it is connected to a small number of adjacent Chls, probably contributes to this, as the necessary uphill transfer to the core antenna likely proceeds through a limited number of pathways with currently unknown energetics. The sensitivity of the electronic structure of Chls to their environment implies that it is important to measure the configuration of the newly formed B31-B32-B33 trimer in PSI in order to validate the structure of this trimer. In particular, B31-B32-B33 is in close proximity to PsaX, a PSI subunit which is absent in *Synechocystis* and this, together with additional sequence changes not included in the

Red_a construct, can potentially affect the configuration of the Red_a loop.

**Cryo-EM structure of the Red_a PSI complex**. We determined the structure of the Red_a PSI trimer using Cryo-EM to a global resolution of 3.1 Å (Table 2 and Supplementary figs. 4 and 5). The majority of the map contains information to 2.4 Å resolution (Supplementary fig. 5) and clearly shows pigment positions and orientations as well as side chains configurations with high confidence (Fig. 6b, Supplementary fig. 6, and Supplementary table 1). Interestingly, we can account for all the Car molecules observed in the 2.5-Å crystal structure in spite of the negative absorption difference in the Car region observed in Fig. 3d, consistent with this signal being associated with a loosely bound Car. As seen in Fig. 6c the added Chl, which is located at the periphery of PSI, is clearly observed. We can also confidently model the loop coordinating this Chl and the conformation closely follows that of the *T. elongatus* PSI structure (Fig. 6c). We conclude that the added B33 Chl adopts the same configuration as in the *T. elongatus* PSI structure with its dipole configuration parallel to B32, which is, in turn parallel to B31. The closest distances between the Chls rings π systems over the trimer is ~3.7 Å, all of these factors contribute to the stronger electronic coupling between them leading to the red-shifted C710 state.

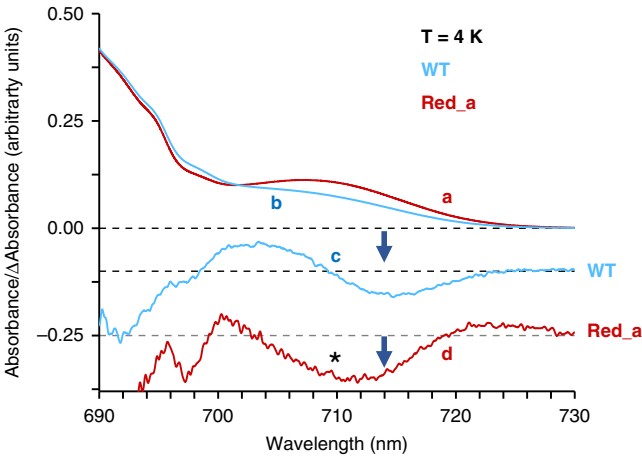

**Fig. 4 B31-B32-B33 Chl trimer is the C710 trap.** Spectra **b** (blue) and **a** (red) show absorption spectra of the WT and Red_a PSI complexes, respectively; recorded at $T = 4$ K, with a spectral resolution of 4 cm$^{-1}$ (see Supplementary fig. 2b for the complete 4 K absorption spectra). The corresponding nonresonant holes (obtained for the burning wavelength ($\lambda_B$) of 670.0 nm) are labeled as curves **c** and **d**, respectively (see text for details). The thick blue arrow marks a broad hole near 714 nm, indicating very efficient EET to the lowest-energy C714 state in WT PSI. The Red_a PSI complex reveals less efficient EET to the C714 state. As a result, the C710 state, indicated by an asterisk, is also bleached.

**Table 1 Assignment of low-energy states to Chls in the PSI core antenna.**

| WT *Synechocystis* | Red_a mutant *Synechocystis* |
| --- | --- |
| C706[a] (B31-B32) | C710[b] (B31-B32-B33) |
| C714[c] (B37-B38) | C714[c] (B37-B38) |
| C707[d] (B7-A31-A32-His 457) | C707[d] (B7-A31-A32-His 457) |

[a]Assigned previously based on single molecule and bulk measurements on PSI complexes from *Synechocystis*[31].
[b]The Red_a mutation induces a spectral shift from 706 to 710 nm (C706 → C710).
[c]C714 trap observed in *Synechocystis*[31].
[d]Assigned to this trimer in refs. [17,74]; though other assignments were also proposed; see refs. [43,44].

Altogether, we observe no additional structural changes between Red_a and the WT PSI, strongly supporting our claim that the addition of B33 is the underlying cause for all of the observed spectroscopic differences.

**Large-scale structural heterogeneity in PSI.** The two PSI structures from *Synechocystis*, the WT trimer, obtained using X-ray crystallography[35] and the Red_a trimer, determined via Cryo-EM in this work, overlap closely over each of their PSI monomers (r.m.s.d of 0.56 Å). We can clearly resolve the 96 Chls (in Red_a), 22 Car, and 9 lipid molecules present in both models. Interestingly, the overlaid trimeric structures revealed some variability between the relative positions of the PSI monomers which is independent of the Red_a mutation. This is illustrated in Fig. 6d which shows both trimers superimposed on each other with one of the monomers (labeled 'A') used for alignment. It is clearly seen that monomers 'B' and 'C' are shifted between the crystal and Cryo-EM structures. We measured the distance between each pairs of monomers and observed that distances between the three PSI monomers vary by as much as 6 Å in the crystal structure. In contrast, inter monomer distances vary by only 0.1 Å in the Cryo-EM structure (Fig. 6d).

These differences suggest that the PSI trimer can exist in a range of conformations which should be observable as heterogeneity in our Cryo-EM dataset. We examined this aspect of the heterogeneity in our dataset using multibody refinement, an implementation of focused refinement with iterative signal subtraction in Relion[36,37]. Multibody refinement consists of defining sub-volumes in the overall map, in our case each PSI monomer was defined as a separate body using low-resolution masks (Supplementary fig. 7a). The orientation of each particle is then optimized to maximize the signal in each sub-volume. Using this strategy, the refinement is free to converge on the different bodies even if their relative positions vary between the different single particle observations. If successful, multibody refinement leads to improved maps and resolution estimates. The variability between the refined positional parameters can be used to measure the heterogeneity and its direction in the particle dataset using principal component analysis[36]. Following multibody refinement of the Red_a dataset, the overall resolution decreased from 3.19 Å to 3. 1 Å (Supplementary fig. 4). The estimated resolution at the peripheral regions of the map improved significantly (Supplementary fig. 7b). Most noticeably, the map around PsaK, a peripheral PSI subunit, improved significantly showing better density around both protein and Chls (Supplementary fig. 7c). Altogether, the improved quality of the map and the appearance of chemically correct features shows that our choice of segmenting the map to three PSI monomers is correct (other segmentation strategies were also tested). We next analyzed the heterogeneity in the dataset using principal component analysis[36].

Supplementary figure 7d shows the eigenvalues of the 18 principal components for the data, where the first two components explain 36% of the total variance in the data. By translating the refined maps along these principal components, it is possible to describe the heterogeneity in the data in the form of movies, where each frame corresponds to 5% of the particle population along the specific component[36]. An example of the particle distribution along the first principle component is shown in Supplementary fig. 7e, the unimodal distribution indicates that this heterogeneity is continuous and does not include distinct states. From Examining the movies generated from the first two principal components (supplementary movies 1–6) showed that they describe very similar forms of heterogeneity depicted as side to side motions of the PSI monomers with the center of PSI, its trimerization domain, serving as a hinge. The similar appearance of two principal components is possible because they involve different PSI monomers described by different parameters, nevertheless the heterogeneity itself clearly belongs to a single class and accounts for more than a third of the variance across the dataset. To better model and measure this heterogeneity, we picked state 1 (representing the tail of the particle distribution along the first principal component containing 15% of the particles population) and state 2 (representing the opposite 15% of the population) and refined the individual PSI monomers into these maps as rigid bodies. The resulting models are shown in Fig. 7a; we measured the distances between the two models at different locations on the PSI trimer. These measurements clearly illustrate that monomers 'B' and 'C' move closer and further away in a circular fashion; the magnitude of the movement is proportional to the distance from the center of the PSI trimer (Fig. 7a). Looking at the PSI trimer from the membrane plane, we measured similar shifts between state 1 and state 2 at the luminal and stromal sides of the membrane, showing that the axis of the movement is nearly parallel to the membrane plane (Fig. 7a bottom and supplementary movies 3 and 4). The overall change in the position of each PSI monomer is ~5 Å at the periphery of the trimer, however, since the monomers move towards and away

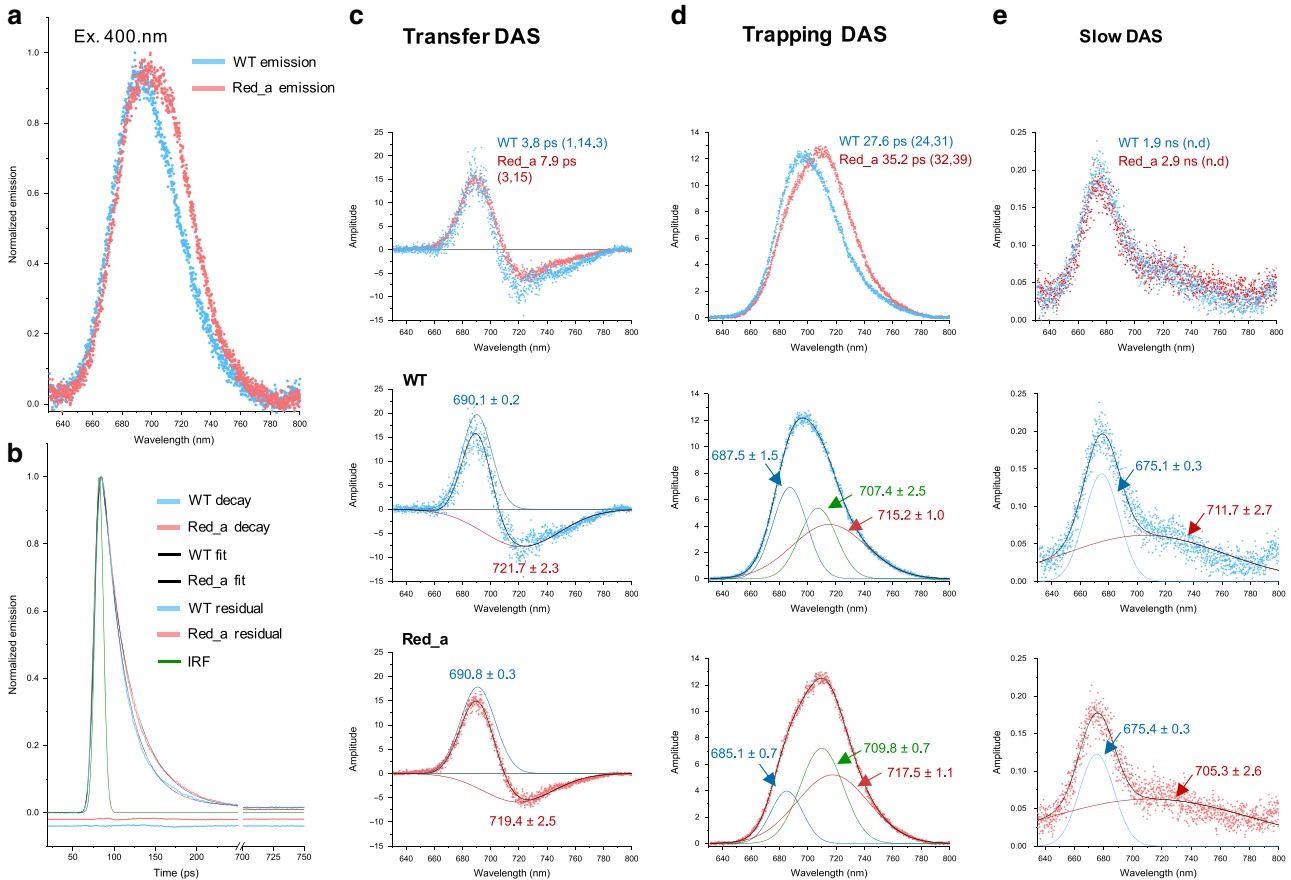

**Fig. 5 Trapping kinetics are altered in Red_a. a** Normalized emission spectra captured by the streak camera summed over the duration of the experimental time window. **b** Normalized decay kinetics summed over 640–800 nm range for WT (blue circles) and Red_a (red circles), IRF (modeled using a Gaussian, fwhm of ~11 ps) is shown in green together with the fit (in black) and residuals for both datasets (shifted by −0.025 and −0.05, for Red_a and WT respectively). Top frames of **c–e** show DAS obtained from global analysis using three decay components. The lifetimes associated with each curve are indicated together with confidence intervals (calculated as a 1% change in $\chi^2$ value of the fit at the indicated parameter value with other parameters treated as free). Middle frames **c–e** show DAS and their Gaussian decomposition for WT PSI. The center wavelengths in nm (±95% confidence intervals) are shown for each component. Bottom frames of **c–e** show DAS and their Gaussian decomposition for Red_a mutant. The center wavelength (±95% confidence intervals) are shown for each component.

from each other, the changes in the distances separating some of the peripheral pigments are larger and can bring some of them to close proximity. This is shown in Fig. 7b, where closest Chl pairs between adjacent monomers in the different states were measured. These Chls are coordinated by the PsaB subunit on one monomer and by the PsaA and PsaK subunits on the other. The average Mg to Mg distance between these clusters in the consensus, averaged, Red_a structure is ~20 Å, significantly larger than the corresponding distances within the core antenna of each monomer which ranges between 9 Å and 13 Å[1,2]. In contrast, at the state representing one conformation (between the 'B' and 'C' monomers) the average distances between these Chls drops to 18 Å, comparable with many antenna – PSI interfaces[38–40]. The Chls configurations at the monomer's interfaces are depicted in Fig. 7c with the distances from the closest state indicated. These distances are consistent with EET between monomers, especially if we consider that they represent an upper bound to 15% of the PSI population. The orientation of one pair in this interface (B9–A20) is unfavorable for EET with an orientation factor value of ~0.08 (Supplementary table 2 lists all the distances and orientation factors for the different states, similar calculation were described in refs. [38,41]). However, other Chl pairs which are closer together are favorably positioned for EET between monomers. We suggest that inter monomer EET through peripheral Chls is

highly likely in a large fraction (approximately a third) of the trimeric PSI population.

To summarize, it is expected that large multimeric membrane complexes will exhibit some structural flexibility, especially when composed of large, complex building blocks, like the PSI monomers. We are able to measure a form of structural heterogeneity in the PSI trimer of cyanobacteria which consists of large-scale shifts in the relative monomer positions. While this heterogeneity is most easily displayed as movies, the time component in these movies cannot be established in Cryo-EM measurements and the time scale of the transitions between the alternative conformations is unknown. Given the high conservation of PSI, we suggest this heterogeneity is a universal feature of PSI trimers.

## Discussion

The existence of red Chls that absorb light in wavelengths longer than P700 in PSI has been known for close to 70 years[42]. The amount and spectral properties of the red Chls vary between different PSI complexes and play a critical role in EET dynamics[10]. By introducing a structural feature into PSI we were able to incorporate a new Chl that contributed to a new red state and extended the absorption spectrum of PSI. The spectroscopic

**Table 2 Cryo-EM data collection, refinement, and validation statistics.**

|  | PSI complex (EMD-20963, PDB-6UZV) |
|---|---|
| *Data collection and processing* |  |
| Calibrated pixel size (Å), Detector, physical pixel size (μm) | 1.05, K2 summit, 5. |
| Voltage (kV) | 300 |
| Total electron dose (e⁻/Å²) | 45 |
| Defocus range (μm) | −1.5 to −3.0 |
| Super pixel size (Å) | 0.525 |
| Symmetry imposed | C1 |
| Initial particle images (no.) | 927,010 |
| Final particle images (no.) | 191,798 |
| Map resolution (Å) | 3.1 |
| FSC threshold | 0.143 |
| Map resolution range (Å) | 2.4–5 |
| *Refinement* |  |
| Initial model used (PDB code) | 5OY0, 6KIG chain K |
| Model resolution (Å) | 3.1 |
| FSC threshold | 0.143 |
| Model resolution range (Å) | 2.4–5 |
| Map sharpening $B$ factor (Å²) | −100 |
| Model composition |  |
| Nonhydrogen atoms | 72,609 |
| Protein residues | 6711 |
| Ligands | 408 |
| $B$ factors (Å²) |  |
| Protein | 8.83 |
| Ligand | 6.22 |
| R.m.s. deviations |  |
| Bond lengths (Å) | 0.009 |
| Bond angles (°) | 0.973 |
| *Validation* |  |
| MolProbity score | 1.86 |
| Clashscore | 11.4 |
| Poor rotamers (%) | 0 |
| Ramachandran plot |  |
| Favored (%) | 96.76 |
| Allowed (%) | 3.22 |
| Disallowed (%) | 0.02 |

changes derived from structural manipulation led to the first identification of a red site in cyanobacterial PSI internal antenna.

Spectral hole-burning and single-complex spectroscopies identified three red states in T. elongatus, C710, C715, and C719[17,31]. Assignment of the B31-B32-B33 Chl trimer to any one of these low-energy states in *T. elongatus* is controversial; for example, some studies assign C719, the lowest-energy state, to this trimer, based on modeling of time resolved data[43], while other studies assign one of the C710/C715 states to this Chl trimer[44,45]. Other studies did not assign a red state to the B31-B32-B33 trimer at all[26,28,30]. In contrast, our results clearly support the assignment of a newly formed C710 state to the B31-B32-B33 trimer in the *Synechocystis* Red_a PSI complex, and by inference suggest that C710 in *T. elongatus* should be assigned to the B31-B32-B33 trimer. This is based on both the differences in absorbance observed in Red_a and the formation of a new hole in the non-photochemical HB spectra at 710 nm observed in Red_a (Figs. 3d and 4). It should be noted that while the conformations of the Chl rings in the newly formed B31-B32-B33 trimer in Red_a are highly similar to the Chl trimer in *T. elongatus*, the Chl coordination of B31 is not identical in both species and was not replaced in Red_a, leaving room for additional differences. Previously, the B31-B32 dimer in the WT *Synechocystis* was assigned the C706 state[31]. This, together with our assignment of the

C710 state to B31-B32-B33, suggest that the low-energy property of this site within the core PSI antenna is conserved and the variation that is observed in red sites reflects modulation of site energy rather than changes in the location of the low-energy sites in PSI.

Recently additional PSI structures became available. In all currently known cyanobacterial PSI structures, there is a correlation between the presence of the Red_a loop, the coordination of a Chl trimer and additional red absorbance. The red loop we added to the *Synechocystis* WT PSI based on *T. elongatus* structure can also be found in *Thermosynechococcus vulcanus*[46], *Fischerella thermalis*[47] and *Nostoc* sp. PCC 7120[48]. The short loop present in *Synechocystis* can be found in *Synechococcus* sp. PCC 7942[49] and *Chroococcidiopsis* sp. TS-821 (PDBID: 6QWJ)[50]. The absence of the red loop in those structures leaves a Chl dimer in this location (which should correspond to the C706 state). The Red_a loop can be found in different forms of PSI, from thermophilic and mesophilic organisms and trimeric and tetrameric oligomeric states of PSI. This loop is a modular chlorophyll coordinator and we suggest that adding this loop to any cyanobacterial PSI lacking this loop will lead to the formation of a B31-B32-B33-like Chl trimer resulting in addition, red shifted, absorption.

The PSI complex from *Synechococcus* WH 7803 was suggested to contains no red Chls[13], Inspecting the PsaB sequence from *Synechococcus* WH 7803 revealed that the B32 coordinating region is different than either *Synechocystis* or *T. elongatus*. Suggesting that the B32 Chl is either displaced or absent, excluding any coupling that exist between B31-B32 in *Synechocystis* and resulting in the absence of the C706 state in the *Synechococcus* WH 7803 PSI complex. We propose a red site assignment in Table 1 which incorporates the findings from the current work and from several other publications (see table for details).

The presence of red Chls in the antenna of PSI necessitates an uphill energy transfer with thermal energy bridging the energetic gap between the red sites and P700 under physiological temperatures. In Gobets et al.[10], a unified minimalistic compartmental model was used to fit kinetic measurements from several cyanobacterial species, concluding that two different red Chl pools, differing in their connectivity to P700, exist in PSI from *T. elongatus* (and *Arthrospira platensis*) compared to a single red pool in *Synechocystis*. The location of the B31-B32-B33 Chl trimer compared to other proposed red sites (see Table 1 and Fig. 8) which are significantly closer to P700, suggest that it makes one pool, separate from other red sites. We suggest that a partial explanation for the relatively large effect of the Red_a mutation on trapping time is attributed to its location. Preliminary calculations suggest that ~4 nm red spectral shift are obtained upon the addition of Chl B33 to the B31-B32 dimer, suggesting that this dimer should be assigned to the C706 state, in agreement with literature data[51]. Thus, at least two red Chls pools exist in this cyanobacterium in both WT and Red_a. Recently, 2DES was applied to PSI from *Synechocystis* and also detected two processes associated with its red Chls, concluding that two red pools exist in this system[51].

It was suggested that red Chls can focus excitation energy to P700, contribute to light harvesting and participate in photoprotection. Clearly, for the B31-B32-B33 Chl trimer focusing excitation energy to P700 is not a plausible suggestion. However, B31-B32-B33 may play a role in EET to PSI from the IsiA antenna. The recent structure of PSI-IsiA shows that the predicted terminal emitters of IsiA are located in close proximity to the Red_a site suggesting a role in EET from IsiA to PSI[38]. The B31-B32-B33 trimer also contributes to light harvesting in the far-red region of the spectrum, in spite of the

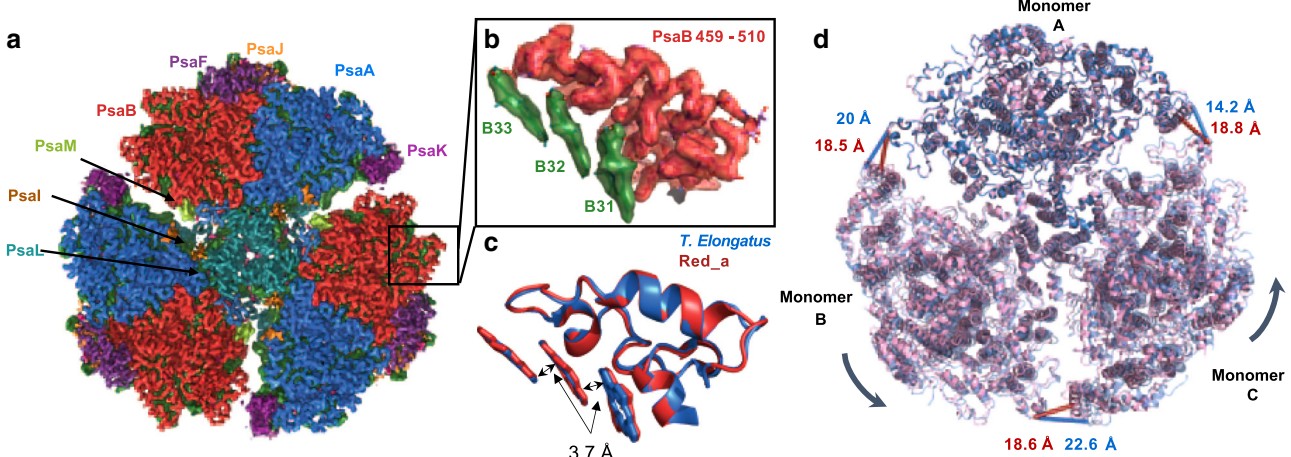

**Fig. 6 Cryo-EM structure of the Red_a PSI complex. a** A view from the lumen side of the membrane of the Red_a PSI trimer map, protein subunits are colored individually, all Chls in green, Cars in pink, and lipids in orange. **b** The map and model of Red_a in the B31-B32-B33 region as viewed from the membrane plane. **c** Superposition of the Red_a structure (in red) and the same region from T. *elongatus* (PDBID: 1JBO, in blue). **d** Superimposing the WT trimer structure (PDBID: 5OY0, in blue, semitransparent in monomers 'B' and 'C') over the Red_a trimer (PDBID: 6UZV, in pink). Monomers 'A' were superimposed using all their protein backbone atoms. A clear shift of ~4 Å is seen for monomers 'B' and 'C'. Distances separating individual PSI monomers are shown in blue for the crystallographic, wild type, model, and in red for the, Red_a, Cryo-EM model, showing very similar monomer spacing in the Cryo-EM model (~18 Å) compared to much larger variation in the crystallographic model (14–22 Å).

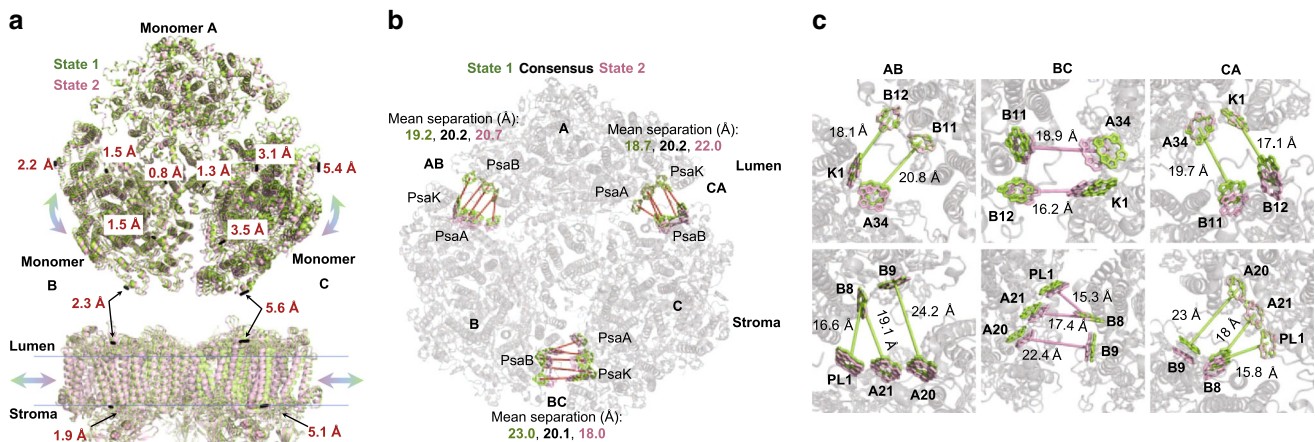

**Fig. 7 Large-scale heterogeneity in PSI. a** Two states, each representing 15% of the particle population are shown from the lumen in green and pink. Monomer A was superposed and several distances between state 1 and 2 are shown as black lines (top). Below is a side view of the same model showing distances measured on opposite sides of the membrane, showing that the movement is nearly parallel to the membrane plane. **b** Two Chl clusters from each state are shown on the background of the Red_a consensus model in gray. The state 1 and 2 configurations of these Chls are shown in green and pink, respectively. The average distances between each cluster for state 1, Red_a and state 2 are shown at each monomer-monomer interface (labeled AB, BC, and CA). **c** The Chl configurations at the three monomer interfaces, on both sides of the membrane, viewed from the luminal side. The distances of the closest state in each interface are shown and colored according the state (state 1 in green and state 2 in pink).

increase in trapping time, the quantum efficiency of PSI remains very high.

It is also possible that B31-B32-B33 Chl trimer participates in photoprotection. The location of Red_a makes a direct interaction with P700 unlikely, however it is interesting to envisage a role of red sites before the assembly of the PsaA-PsaB heterodimer and the formation of P700 itself. In the whole complex, P700 functions as a quencher of the PSI Chls, yet in the assembly process of PSI, PsaB (in *C. reinhardtii*, for example) is synthesized first and integrated into the thylakoid membrane before PsaA[52]. The assembly process of PSI in cyanobacteria is still unclear, however there is evidence for separate transcripts for PsaA and PsaB in addition to a whole transcript of PsaAB in *Synechocystis* implying the possibility of similar assembly mechanism as in *C.*

*reinhardtii*[53,54]. P700 is created only upon dimerization of PsaA and PsaB and the formation of the whole internal electron transport chain of PSI is completed with the subsequent addition of the stromal subunit, PsaC. PsaA and PsaB already carry Chls in their internal antenna that are prone to damage without the presence of the energy quencher, P700. In this case, red Chls can be a part of a mechanism for energy quenching to prevent Chls triplet formation and damage to each subunit. The reason that we find this red site on PsaB is due to the role of PsaB as an anchor for PsaA during PSI assembly. Each PSI monomer coordinates 22 Car molecules across the antenna which are known to play a crucial role in photoprotection. These Cars interact with approximately two thirds of the antenna Chls, close examination shows that a Car is located in the vicinity of each of the predicted

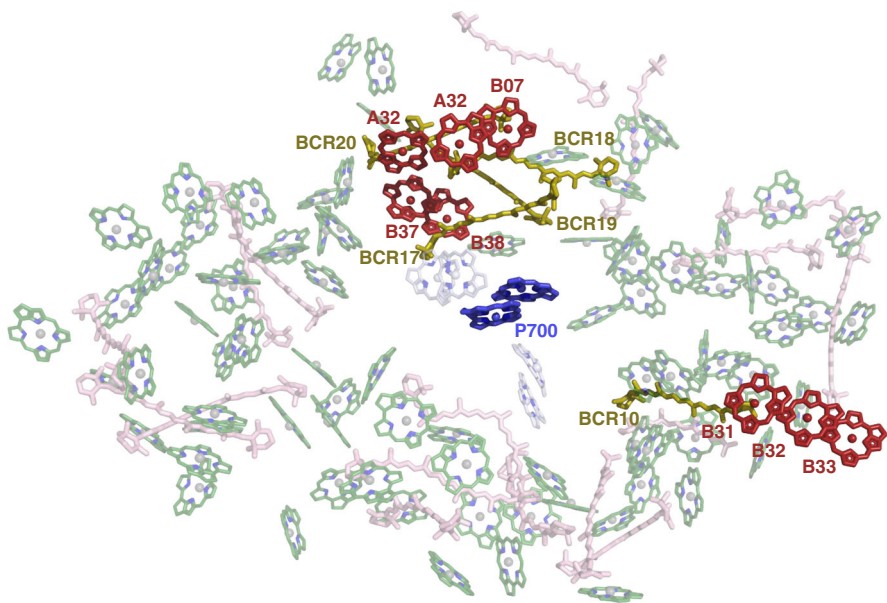

**Fig. 8 Carotenoids and proposed red sites configuration in PSI.** The structure of PSI with the core antenna Chls (green) and Cars (pink) shown in the background, predicted red Chls are in red and β-carotenes in the vicinity of the predicted red Chls are in olive, P700 Chls and the electron transport chain accessory Chls are in blue and gray, respectively.

red sites in PSI (Fig. 8). The close interaction between red Chls and Cars may be an important mechanism for photoprotection in PSI.

The large-scale heterogeneity we observe occurs in the trimeric form of PSI and is independent of the Red_a mutation. Previously, the red states in the cyanobacterial PSI were also shown to change significantly between trimeric and monomeric PSI forms, with red absorption significantly decreased in monomers compared to trimers[32,55]. The reasons for these changes can be either conformational changes that modulate site energy or loss of pigments that occur upon monomerization. We modeled the heterogeneity in PSI in the form of rigid monomers which cannot detect any conformational changes within the PSI monomers. However, it seems very reasonable that conformational changes occur as part of the different states the monomers occupy, the extent of these conformational changes could affect the energy of a red site. One clear candidate for this is the site formed by the B7-A31-A32 cluster assigned to the C707 state (see Table 1 and Fig. 8), which is in close proximity to the PsaL subunit known to be essential for trimerization. Another clear consequence of the observed states are the large changes in distances between two groups of Chls located at the "corners" of PSI (Fig. 7). These changes enable EET between the different PSI monomers.

One of the proposed functions of the trimeric organization of PSI in cyanobacteria is to enable EET between monomers in the trimers. This was suggested to relieve excess energy burden from one monomer by the adjacent monomer and slower rates of P700 oxidation were observed in the trimer of *Arthrospira platensis* in comparison to the monomer, suggesting energy migration between monomers[56]. The very same group of peripheral Chls highlighted in Fig. 7 was proposed to mediate this EET, however, large scale changes in monomer packaging had to be assumed[57]. The heterogeneity identified in the current study means that a substantial portion of the trimeric PSI population occupies positions compatible with monomer to monomer EET, although the nature of the heterogeneity is very different than the one proposed in[57].

PSI is one of the essential components of oxygenic photosynthesis. The high density of pigments in the core of PSI presents a great challenge to deciphering the structural determinants

of its electronic structure. The identification of the B31-B32-B33 trimer as one of the red states brings us a step closer to understanding the design rules that drive the evolution of photosystems and allow these assemblies to tune their light-harvesting properties.

## Methods

**Strain construction and growth**. *Red_a strain construction*: a plasmid (p60) containing the entire PsaAB operon marked with Kanamycin resistance gene 0.8Kb downstream to PsaB was constructed from four PCR amplified fragments. The PsaAB operon and PsaB downstream fragments were amplified using the PsaAB_fwd/PsaAB_rev and PsaBdown_fwd/PsaBdown_rev primer pairs from the *Synechocystis* sp. PCC6803 genome. The pA15 Origin of replication was amplified from pACYC184 using primers pA15ori_fwd and pA15ori_rev. The kanamycin resistance gene was amplified from pET30 using Kan_fwd and Kan_rev. The four fragments were assembled using the NEBuilder® HiFi DNA Assembly Master Mix. The Red_a mutation was constructed into p60 by adding the red loop sequence using the Red_a fragment1 and fragment2 primers and p60 as a template. The two fragments were assembled using the NEBuilder® HiFi DNA Assembly Master Mix. All plasmids were sequenced before being used to transform *Synechocystis* sp. PCC6803 according to standard protocols. Complete and correct replacement of PsaB was verified by PCR and sequencing. The primers used to construct the Red_a mutant are listed in supplementary table 3.

*Culture conditions*: cyanobacteria were cultured in BG11 medium supplemented with 6 μg/ml Ferric ammonium citrate, 5 mM glucose, and 10 μg/ml Kanamycin under continuous white light (~40 μE) in 30 °C.

*Growth tests*: wild type, mutant, and control cells (containing the resistance gene but not the mutation) were cultured in BG11 liquid medium supplemented with 6 μg/ml Ferric ammonium citrate and 5 mM glucose under continuous white light (~40 μE) in 30 °C. The optical density was adjusted to 0.5 at 730 nm. Each culture was diluted x5, x25, x125, and x625 and grew on BG11 plates in different light intensities.

*Photosystem I purification*: 20 l of cells were grown in BG11 (supplemented with 6 μg/ml ferric ammonium citrate and 5 mM glucose, plus 10 μg/ml kanamycin for the red strain) under light intensity of ~40 μE at 30 °C. Cells were harvested using centrifugation in a F9 6 × 1000 LEX rotor in Sorval LYNX 6000 centrifuge (Thermo Scientific) for 5 min, 4700 × *g*, all further sample handling was done on ice. Cells were washed once using ~200 ml of STN1 buffer (30 mM Tricine-NaOH pH 8, 15 mM NaCl, and 0.4 M sucrose). Spun down again using a F20-12 × 50 LEX rotor, 5 min, 12,857 × *g*. Finally, cells were resuspended in 50 ml of STN1, and broken by Constant Systems cell disruptor (two cycles at 30 kpsi). The lysate was cleared by centrifugation in a F20-12 × 50 LEX rotor for 10 min at 18,514 × *g*. Membranes in the supernatant were pelleted using ultracentrifugation (Ti70 rotor, 207,870 × *g* for 2 h), and resuspended in 50 ml STN2 (30 mM Tricine-NaOH pH 8, 150 mM NaCl, 0.4 M sucrose). After resuspension in STN2, the membranes were incubated on ice for 30 min, then collected again (Ti70 rotor, 207,870 × *g*, 2 h), and resuspended in

~15 ml of STN1. Chlorophyll concentration was determined according to Amon[58]. n-Dodecyl β-ᴅ-maltoside (DDM, Glycon) was added to the membranes at a 15:1 DDM to chlorophyll mass ratio. The suspension was gently mixed by hand a few times then incubated on ice for 30 min. After solubilization, the insoluble material was discarded using ultracentrifugation (Ti70, 207,870 × $g$, 30 min). The solubilized membranes were loaded onto a DEAE gravity flow column (Toyopearl C650). The complexes were eluted using a linear NaCl gradient (15–350 mM NaCl) in 30 mM Tricine-NaOH pH 8, 0.2% DDM. Dark green fractions were collected, PEG3350 (Hampton research) was added to a final concentration of 8.5% and the PSI trimer were precipitated using a 5 min, 3214 × $g$ spin in a F20-12 × 50 LEX rotor. The green precipitate was resuspended in 30 mM Tricine-NaOH, pH 8, 0.05% DDM, 75 mM NaCl, and loaded onto a 10–30% sucrose density gradient, prepared in 30 mM Tricine-NaOH pH 8, 0.05% DDM, 75 mM NaCl. Following centrifugation (SW40 rotor, 242,832 × $g$, 12 h), the PSI trimer was collected and precipitated with 8.5% PEG3350 at 150 mM NaCl. After a centrifugation step (18,407 × $g$, 5 min) the green precipitate was resuspended in 30 mM Tricine-NaOH, pH 8, 0.05% DDM, 75 mM NaCl, and loaded onto a second 10–30% sucrose density gradient done in an SW60 rotor (392,960 × $g$, 4 h). The green band was collected and used for subsequent experiments. For the experiments using a third sucrose gradient (Supplementary fig. 1), the sample was diluted to 50% using 30 mM Tricine-NaOH, pH 8, 0.05% DDM, 75 mM NaCl, and loaded on a 10–30% sucrose density gradient prepared as before (SW40 rotor, 242,832 × $g$, 12 h).

**Absorption and fluorescence spectroscopy**. *Room temperature measurements*: absorption spectra were recorded on a Cary 4000 UV–Vis spectrophotometer (Agilent Technologies). Fluorescence spectra were recorded on a Fluoromax-4 spectrofluorometer (HORIBA Jobin-Yvon). Samples were diluted to an optical density of 0.8 and 0.1 at 680 nm for absorption and fluorescence measurements respectively, using buffer containing 30 mM Tricine-NaOH pH 8, 15 mM NaCl, and 0.05% DDM. The resulting spectra were normalized to the area of the chlorophyll Q bands (550–750 nm) using 95 chlorophylls for the wild type and 96 chlorophylls for Red_a. For 77 K fluorescence measurement samples were adjusted to an OD680 of 0.1 in a buffer of 50% glycerol 30 mM Tricine pH = 8.0, 15 mM NaCl, and 0.02% DDM. An Oxford instruments Cryostat was used to cool the sample to 77 K (cells were plunged into liquid nitrogen and measured immersed in liquid nitrogen). Figures were prepared using OriginPro (OriginLab).

*Low-temperature absorbance*: for low-temperature experiments, samples were diluted with 1:2 (v/v) buffer:glass solution. The glass forming solution was 55:45 (v/v) glycerol:ethylene glycol. Briefly, a Bruker HR125 Fourier transform spectrometer was used to measure the absorption and nonresonant hole-burned (NRHB) spectra with resolution of 4 cm⁻¹. In all, 488.0 nm excitation for NRHB spectra was produced from a Coherent Innova 200 argon ion laser. In total, 670.0 nm excitation was obtained from a Coherent CR899 ring dye laser pumped by a Spectra-Physics Millennia Xs diode laser (532 nm). With laser dye LD 688 (Exciton), laser excitation in the spectral range of 650–720 nm (for resonantly burned holes) was available with a line width 0.07 cm⁻¹. Laser power in all experiments was precisely set by a continuously adjustable neutral density filter. Low temperature (4 K) experiments were performed using an Oxford Instruments Optistat CF2 cryostat with sample temperature read and controlled by a Mercury iTC temperature controller. The hole-burning (HB) spectroscopy relies on differences observed in the absorption spectrum of a low-temperature system after narrow-band laser excitation. If a pigment molecule (in resonance with the laser) experiences photochemical reaction, it ceases to absorb at its original wavelength/frequency and one speaks of photochemical HB (PHB). In this work, however, we use non-photochemical HB (NPHB); in this case, the pigment molecule does not undergo a chemical reaction, but its immediate environment experiences rearrangement on the protein energy landscape[3,4]. NPHB may result in the formation of persistent holes, meaning the holes persist after the initial excitation is turned off, as long as low temperature is maintained. (The HB spectrum is obtained as the difference between the measured absorption spectrum before and after laser excitation). The key information provided by HB spectroscopy (relevant to data discussed in this work) includes lifetimes of the zero-point level of S₁(Q_y)-states due to EET, as determined by the widths of zero-phonon holes (ZPHs) and/or electron-phonon (protein) coupling parameters.

*Time resolved fluorescence decay*: samples were diluted to an optical density (680 nm) of 0.75 in 30 mM Tricine pH 8.0, 15 mM NaCl, 0.05% DDM, 5 mM Ascorbic acid, and 1.5 μM phenazine methosulfate. The excitation beam was set to excite the sample as close as possible to the outer edge of the cuvette to reduce reabsorption effects. Experiments were carried out at room temperature in a 10.00 × 10.00 mm quartz cuvette (Hellma Analytics) continuously mixed using a magnetic stirrer. Sample were illuminated by 100 fs pulses from a mode-locked Ti:S laser (Mira 900, Coherent Laser Inc., Santa Clara, CA) pumped by a frequency-doubled Nd:YVO 4 laser (44% from an 18 W Verdi, Coherent Laser Inc.). The repetition rate was reduced to 4.75 MHz using a pulse picker (Model 9200, Coherent Laser Inc.) and the excitation light was frequency-doubled to 400 nm and focused on the sample cuvette. To avoid singlet-singlet annihilation, the pulse energy was reduced to ~0.1 nJ using a neutral density filter. Fluorescence was collected at a right angle to the excitation beam and focused on the entrance slit of a Chromex 250IS spectrograph coupled to a Hamamatsu C5680 streak camera with a M56757 sweep unit. Time-intensity surfaces were recorded at a time scale of 0.8 ns on a Hamamatsu C4742 CCD camera. The fwhm of the overall instrument response function (IRF) was ~12 ps at the 0.8 ns timescale. Global analysis was carried out in MATLAB using a set of scripts available from the author upon request following the methodologies described in[33,34].

**Cryogenic electron microscopy**. *Sample preparation for single particle Cryo-EM analysis*: the PSI band from the sucrose gradient was collected, NaCl was added to a final concentration of 150 mM and the complex was precipitated using 8.5% PEG3350. After centrifugation (2,300 × $g$, 5 min in an Eppendorf tabletop), the green precipitate was resuspended in buffer (30 mM Tricine-NaOH pH 8, 150 mM NaCl 0.02% DDM), and any undissolved material was removed by repeating the centrifugation step (18,407 × $g$, 5 min). The chlorophyll concentration in the soluble material was adjusted to 1 mg/ml using the above buffer. A 3 μl of The PSI complex was added to a holey carbon grids (C-flat 1.2/1.3 Cu 400 mesh grids (Protochips, Raleigh, NC)) after soaking the grids in buffer. The sample was vitrified by flash plunging the grid into liquid ethane using manual plunger with blotting time of ~6 s. The grids were stored in liquid nitrogen until data collection.

*Data acquisition*: the Cryo-EM specimens were imaged on a Titan Krios transmission electron microscope (Thermo Fisher - FEI, Hillsboro, OR). The electron images were recorded using a K2 Summit direct electron detect (DED) camera (Gatan, Pleasanton, CA) at super-resolution counting mode. Automated data collection was with SerialEM[59] with stage shift in between exposures. The defocus was set to vary between −1.5 and −3 μm. The super-resolution pixel size was 0.525 Å at the specimen level corresponding to ×47,600 magnification. The counting rate was adjusted to 5.6 e⁻/Å²sec. Total exposure time was 8 s, fractionated into 40 frames, accumulating to a dose of 44.8 e⁻/Å².

*Data processing*: a flow chart describing data handling is shown in Supplementary fig. 4. MotionCor2[60] was used to register the translation of each sub-frame, and the generated averages were binned 2X and dose-weighted[61]. Contrast Transfer Function (CTF) parameters for each corrected micrograph were determined using CTFFIND4[62]. Relion3.1 was used for the subsequent data processing[37]. A set of manually picked particles (~400) was subjected to a few rounds of unsupervised 2D classification. Four class averages, which represented different orientations of the expected particle, were selected and used as the templates for the automated particle picking procedure as implemented in Relion which yielded 927,010 particles. This particle set was subjected to several rounds of unsupervised 2D classification leading to a set of 231,381 particles which were extracted as boxes of 310 pixels with a final pixel size of 1.05 Å/Px. 3D reconstruction using this set led to a map with an overall resolution of 3.9 Å after masking. This particle set was subjected to 3D classification using six classes with most of the particles classified in clear PSI trimer classes and minor fraction classified into a "junk" class, the trimeric classes were grouped together and refined again and then the 3D classification was rerun (three cycles of classification followed by refinement) until only trimer classes were obtained. All classes were judged to be similar by visual inspection of the maps and all refined to comparable resolution (~4 Å). At this point all the classes were combined to yield a 191,798 particle set. Per particle CTF refinement was carried out using relion3.1[63] followed by Bayesian polishing[64]. The final resolution of this particle set refined in C1 was 3.35 Å. After this refinement the particle set was symmetry expanded using the C3 group and refined to a final resolution of 3.19 Å. Finally, multibody refinement[36] using each monomer as a separate body yielded the final map at a resolution of 3.1 Å according to gold standard FSC criteria[65]. The final map was sharpened using the post processing procedure in Relion, the b-factor used for sharpening was −100 Å². Local resolution was estimated using ResMap[66]. The three PSI monomers were refined into each of the three maps as rigid bodies and a composite map was generated in phenix[67] using the combine_focused_maps tool.

*Model building and refinement*: the initial PSI model was taken from the 2.5-Å x-ray structure of the trimeric PSI from *Synechocystis* (PDBID: 5OY0)[35] with PsaK taken from[68] (PDBID: 6KIG) and the sequence was manually changed to the *Synechocystis* sequence. The model was docked into the map using PHENIX. The final model was refined against the Cryo-EM density map using phenix. real_space_refine[69,70]. Final model statistics are shown in Table 1. Map model fitting and resolution was also checked using MapQ, a USCF Chimera plugin[71]. PyMOL[72] and UCSF Chimera[73] were used to generate all images. Dipole orientations were calculated using an R script similarly to[38,41].

**Reporting summary**. Further information on research design is available in the Nature Research Reporting Summary linked to this article.

## Data availability
The final model (PDB 6UZV) and map (EMD 20963) were deposited in the Protein Databank and Electron Microscopy Database, respectively. All other data are available from the authors upon reasonable request. Source data are provided with this paper.

## Code availability
Global analysis of fluorescence data was carried out using a MATLAB script, dipole orientations values were calculated using an R script, both available from the author upon reasonable request.

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

## Acknowledgements

We would like to thank R. Blankenship for critical reading of the manuscript. We would like to acknowledge the use of the Titan Krios at the Erying Materials Center at Arizona State University and the funding of this instrument by NSF MRI 1531991. This study is funded by a startup grant from Arizona State University (to Y.M). The work performed at Kansas State University was supported by the Chemical Sciences, Geosciences and Biosciences Division, Office of Basic Energy Sciences, Office of Science, U.S. Department of Energy (Grant No. DE-SC0006678 to R.J.).

## Author contributions

Y.M., H.T., R.J., and A.K. performed experiments, analyzed data, and wrote the manuscript. Z.D., R.R., S.L., and D.W. performed experiments.

## Competing interests

The authors declare no competing interests.
