## [Peer Review File · Nature Communications]

REVIEWER COMMENTS

Reviewer #1 (Remarks to the Author):

The authors introduced mutations to clarify the location of red Chl in photosystem I based on the structure between *Synechocystis* sp. PCC6803 and *Thermosynechococcus elongatus*. Based on the structural comparison between two, it is easy to imagine that the loop portion of the protein contributes to the Chl trimer. As a result, the presence of red Chl has indeed been confirmed by the mutation. The authors compare only the PSI of *Synechocystis* sp. PCC6803 and *T. elongatus* but structures of other cyanobacterial PSI have also been reported. Please describe how this mutation would play out in other structures. And it is true that photosystem I heterogeneity exists in trimer structure, but it is difficult to understand the relationship between photosystem I heterogeneity and red-Chl triplet (B21-B22-B23). Please explain more clearly the direct relationship between the heterogeneity and red-Chl triplet.

The description of the PSI purification is short. The method section should be described in more details. I suggest the authors to provide an additional ED figure to show the characterization of PSI samples, including the profiles of chromatography and 1st sucrose density gradient centrifugation. In that case, I believe you can also isolate the monomer PSI. Was red-Chl also conserved in monomeric photosystem I?

Please explain that the long-wavelength component of the fluorescence is not a vibrational band. In particular, the low-energy band in Figure 5e looks like a vibrational band.

Red_a2 cells appear to fade in high light treatment (550 μ E). What is the difference between Red_a1 and a2?

Page 5, line 22: What is the CT states? Please explain the abbreviations in the first appearance.

Show the entire visible absorption spectrum (350-750 nm) at 4K. The Soret and Qy maximum were same between two?

In Fig. 3b, PsaA/B, the intensity of the PsaA/B band is lighter in the wild type. Why is this? What is the band that is higher molecular weight than PsaA/B?

In Fig. 3f, the authors described that some additional Car are present in the WT preparation. However, the carotenoid content was same in Cryo-EM structure. Is this difference in the absorption spectrum an error? From the molar absorption coefficient of the pigment and the difference in absorbance, it should be possible to determine how many molecules of the pigment have changed.

Data not shown is present in several places.

Based on the Journal policy, all data are shown either in the main text or the Supplementary Information.

Reviewer #2 (Remarks to the Author):

The manuscript by Toporik et al. reports the structural and spectroscopic analysis of the red_a PSI complex, a chimeric PSI from *Synechocystis* PCC 6803, in which one additional chlorophyll B33 was introduced. The addition of B33 results in a Chl trimer B31-B32-B33 that is absent in the wild type PSI complex of *Synechocystis* PCC 6803. The authors compared the spectroscopic properties of the red_a with the wild type PSI, and showed that the newly formed Chl trimer results in the

red shift spectra and creates a new red site c710. They also solved the cryo-EM structure of the red_a PSI complex to show that it has highly similar structure as the wild type PSI, indicating that the formation of c710 is due to the presence of the Chl trimer B31-B32-B33. In addition, the authors found a large-scale structural heterogeneity of the trimeric PSI, suggesting the movement of PSI monomers within a trimer.

This is a quite interesting work, and the first one to identify the red site in the PSI core, thus advances our knowledge. However, I have a major concern about the quality of the cryo-EM structure. The model-map fit is very poor, even for the core subunits PsaA and PsaB, as shown in the PDB validation report. It seems something wrong with the coordinate or the validation report. I require the authors to further adjust and refine their structure and validate it again when revise their manuscript. Since the authors discuss about the structural heterogeneity and the short distances between interfacial chlorophylls, the accurate structural model should be provided.

Minor points

- I suggest that the authors to comment whether the PSI complex from *T. elongatus* shows the same spectroscopic property of c710, since the *T. elongatus* PSI has the same Chl trimer.
- Page 11, the citation of "Supplementary figure 5" is probably wrong. I think it should be "Supplementary figure 6".

Reviewer #3 (Remarks to the Author):

This manuscript utilized the mutagenesis approach to identify the location of a red site in the core antenna of cyanobacteria photosystem I. The authors utilized spectrometry and cryo-electron microscopy to validate their finding. I will leave other reviewers to qualify the scientific novelty and merit of this work. I only give some comments to the issue of cryoEM image processing and molecular modeling.

1. According to the validation report submitted from the authors, it seems that the current model has a very good geometry score but very poor clash score. And the reported fitness between model and EM map is rather poor with all most residues appear poor fit. This issue should be fixed. I guess there could be at least three points to be considered. First, the authors should think about the correctness of pixel size. Was the pixel size of the camera under the condition of data collection well calibrated using one standard specimen? The authors could also use some software packages to check this problem. Second, the Ramachandran statistics and geometry of bond and angle are rather good. It seems that the authors utilized a tight geometry restriction during model refinement or performed a further round of geometry optimization after model refinement. This would induce a poor fitness of cryoEM map and is not reasonable. Third, the final cryoEM map was post-processed with a proper mask and b-factor. The authors should check whether the mask is too tight and thus some weak densities were truncated.

With the above issue solved, the authors are also suggested to calculate and report the Q score of their model (chain by chain) according to the recent literature Nature methods 17, 328-334. Of course, a new validate report is needed in the next round of review.

2. The authors should provide more detailed information about cryoEM data collection. For example, what kind of software was used to collect data automatically? Whether the beam/image shift protocol (or just stage shift) was applied during data collection? How many frames were fractionated in each exposure movie? The representative micrograph with few number of particles in S4a seems a cropped version of the original one. The authors should present the representative original one. The description of magnification and pixel size in Methods would be problematic (P22, line 18). The nominal magnification should be like 47,600X. The accuracy of pixel size should not be such high with four digits. The description of counting rate (P22, line 19) is not correct. It should be dose rate. Again, the four digits of dose rate is not rigorous.

3. The data processing procedure should be also revised carefully with more information provided. Which version of RELION was used? The citation of RELION was missed. After Fourier-crop, the pixel size has been doubled. Thus the pixel size of 1.05 Å/Px is not yielded from particle extracting in RELION. Whether the CTF refinement (which is essential to further push resolution) was performed? Whether the procedure of particle polishing was performed? The b-factor should have a unit of Å². The authors are suggested to validate the orientation distribution effect by calculating Eod score according to the recent literature Nat Commun 8, 629.

Dear Referees,

We are resubmitting our manuscript entitled “The structure of a red-shifted Photosystem I reveals the location of a red site in the core antenna.” for consideration in Nature communications. We would like to thank all three reviewers for their constructive criticism which improved the quality of the work. We made every effort to describe sample preparation, data collection and image processing as fully and openly as possible. We have reanalyzed our dataset and now present a map with improved resolution of 3.1 Å. We have refined our model into this map and can now present a much-improved correlation between the model and map which extends to 2.8 Å at FSC of 0.143. Importantly, the heterogeneity we describe in the paper is still present in the current analysis, the proportion of the population occupying extreme positions decreased to some extent and we have indicated all the changes in the manuscript file in red. Below is our point by point response to all of the comments made by the reviewers, we’ve highlighted in bold the main points raised by the reviewers and our answers are in red.

Respectfully,

Yuval Mazor, PhD.

Reviewer #1 (Remarks to the Author):

The authors introduced mutations to clarify the location of red Chl in photosystem I based on the structure between *Synechocystis* sp. PCC6803 and *Thermosynechococcus elongatus*. Based on the structural comparison between two, it is easy to imagine that the loop portion of the protein contributes to the Chl trimer. As a result, the presence of red Chl has indeed been confirmed by the mutation. The authors compare only the PSI of *Synechocystis* sp. PCC6803 and *T. elongatus* but structures of other cyanobacterial PSI have also been reported. **Please describe how this mutation would play out in other structures.**

The research question that initiated this contribution was how and what structural features effect the spectroscopic properties of PSI complexes and especially the red chlorophylls content of PSI complexes. It was already known that *Synechocystis* sp. PCC6803 and *Thermosynechococcus elongatus* contain different amounts of red

chlorophylls that vary in their wavelengths. However, until 2014, the only available cyanobacterial PSI structure was from *Thermosynechococcus elongatus*. Hence, when the structure of *Synechocystis* sp. PCC6803 was solved by Mazor et al. it enabled structural comparison in order to explore the structural features that determine the spectroscopic properties unique to PSI complexes from different cyanobacteria species. More recently additional PSI structures became available. In all currently known cyanobacterial PSI structures, there is a correlation between the presence of this loop, the coordination of a chl trimer and the presence of additional red absorption. The red loop we added to the *Synechocystis* sp. PCC6803 WT PSI based on *Thermosynechococcus elongatus* structure can also be found in *Thermosynechococcus vulcanus*¹, *Fischerella thermalis*² and *Nostoc* sp. PCC 7120³. The short loop present in *Synechocystis* sp. PCC6803 can be found in *Synechococcus* sp. PCC 7942⁴ and *Chroococcidiopsis* sp. TS-821 (PDBID: 6QWJ)⁵. The absence of the red loop in those structures leaves a chl dimer in this location. The loop can be found in different forms of PSI, from thermophilic and mesophilic organisms and trimeric and tetrameric forms of PSI. This loop is a modular chlorophyll coordinator that is present or absent depending to the species. We assume that adding this loop to any cyanobacteria lacking this loop will lead to addition of a third chlorophyll coupled to the dimer resulting in red shift absorption spectrum and all current observations support this. We have added a similar paragraph to our discussion.

it is true that photosystem I heterogeneity exists in trimer structure, but it is difficult to understand the relationship between photosystem I **heterogeneity and red-Chl triplet (B21-B22-B23). Please explain more clearly the direct relationship between the heterogeneity and red-Chl triplet.**

The heterogeneity in the trimeric structure of PSI was discovered during the processing of the red PSI data and the use of multibody refinement procedure. We suggest that this heterogeneity is a universal property of all trimeric PSI structures and is independent of the red chl content of the specific PSI complex. We observe similar heterogeneity in three PSI trimers from different cyanobacterial species in our lab with very different red sites content and different configurations of the Red_a region. We have indicated clearly in the manuscript that we don't think this heterogeneity is related to the B31-B32-B33 Chl trimer.

The description of the PSI purification is short. The method section should be described in more details. I suggest the authors to provide an additional ED figure to show the characterization of PSI samples, including

the profiles of chromatography and 1st sucrose density gradient centrifugation. In that case, I believe you can also isolate the monomer PSI. **Was red-Chl also conserved in monomeric photosystem I?**

We have expanded our description of the PSI purification method and added an image of the first sucrose gradient (Supplementary figure 1). We utilize gravity flow to run our ion exchange column and rely on visual inspection of the chl fractions to identify the major Chl peak, for these reasons we don't have the chromatography profile requested by the reviewer. We did observe the added red chlorophyll on absorbance measurements done on monomeric PSI, these results are complicated due to the presence of variable amounts of PSII contaminations in the monomeric PSI fraction. We are working towards identifying the underlying reason for the red absorbance differences between monomers and trimers as part of another study. According to our current findings and what we would expect from the structure, Red_a is not involved in these differences, the difference in absorbance due to Red_a is maintained in PSI monomers, however this is work in progress.

Please explain that the long-wavelength component of the fluorescence is not a vibrational band. In particular, the low-energy band in Figure 5e looks like a vibrational band.

The low amplitude (1/600 the amplitude of the transfer component), slow DAS component is attributed to free pigments in most publications. We don't think that its low energy band is vibrational in nature mainly due to its close proximity to the main emission band at 675 nm which is different from the relative location of the vibrational band in the absorption spectra of free chlorophyll. At the same time, we don't think this component is important for our data and its low amplitude reflects the low concentration of free pigments in our preparations, in addition to being similar between wild type and Red_a trimers. For these reasons we prefer to minimize our discussion of this component and merely state its presence, and this is the common way this issue is addressed in similar publications.

Red_a2 cells appear to fade in high light treatment (550 uE). What is the difference between Red_a1 and a2?

Red_a1 and Red_a2 are strains that originated from two different colonies from the same transformation experiment and represent independent transformation events. Both strains contain the resistance gene and the Red_a mutation and all the spectroscopic findings at the cellular level were duplicated in both strains to control for any variability between cells. The difference the reviewer is referring to

has to do with experimental variability, which is not completely eliminated, we concluded that there is no growth difference between wild type and Red_a cells under high light based on numerous experiments.

Page 5, line 22: What is the CT states? Please explain the abbreviations in the first appearance.

We have added “charge transfer” after the CT abbreviation in page 5 line 22.

Show the entire visible absorption spectrum (350-750 nm) at 4K. The Soret and Qy maximum were same between two?

The absorbance data collected at 4 K was collected using a high resolution Bruker HR125 Fourier transform spectrometer at a range of 500-1000 nm. We present the complete spectrum in the supplementary material (Supplementary figure 2b). It is clearly seen that the major difference is located at the chlorophyll Qy.

In Fig. 3b, PsaA/B, the intensity of the PsaA/B band is lighter in the wild type. Why is this? What is the band that is higher molecular weight than PsaA/B?

To prevent aggregation, which occurs in the case of very hydrophobic proteins, we do not boil our samples prior to SDS-PAGE analysis. As a result, disassociation is sometimes partial, and this is the reason for the high molecular weight band (most probably a PsaAB dimer) and also for the changes in band intensities. These changes do not reflect changes in subunit stoichiometry, as we would have seen these in our Cryo-EM analysis.

In Fig. 3f, the authors described that some additional Car are present in the WT preparation. However, the carotenoid content was same in Cryo-EM structure. **Is this difference in the absorption spectrum an error? From the molar absorption coefficient of the pigment and the difference in absorbance, it should be possible to determine how many molecules of the pigment have changed.**

We don't think that the difference in the absorbance spectrum is an error. In measurements done on pigments extracted from wild type and Red_a trimer using 80% acetone (to control for any solvent effects) we observe a negative difference in Red_a which corresponds to a one carotene molecule per 96 chlorophylls. This suggests to us that the wild type complex binds one more beta carotene molecule than the Red_a complex under our purification conditions.

Figure 1: a. Spectra measured on pigments extracted from both trimers in 80% acetone. b. Difference of the two spectra in 'a'.

The fact that we do not observe a “missing” carotene in the Red_a structure may be due to the fact that this beta carotene is missing in the wild type structure. This may be a peripheral carotene loosely associated with PSI which may explain why it is not modeled in structures. We prefer not to speculate on the nature of the binding site even though it seems to be related to the Red_a loop, since we did not determine the WT trimer structure. To show that this extra carotene is indeed loosely bound to PSI, we run our samples through an additional sucrose gradient (without any additional chromatography steps) and remeasured the absorbance of both WT and Red_a trimers. In this spectrum the negative difference in the carotene region is completely gone while the difference in the Qy region are preserved (supplementary figure 1 c and d). We still maintain, based on our experience, that the best PSI sample comes from the second gradient and this is also the most appropriate comparison to the crystal structure.

Data not shown is present in several places. Based on the Journal policy, all data are shown either in the main text or the Supplementary Information.

We removed all the remarks referring to data not shown.

Reviewer #2 (Remarks to the Author):

The manuscript by Toporik et al. reports the structural and spectroscopic analysis of the red_a PSI complex, a chimeric PSI from *Synechocystis* PCC 6803, in which one additional chlorophyll B33 was introduced. The addition of B33 results in a Chl trimer B31-B32-B33 that is absent in the wild type PSI complex of *Synechocystis* PCC 6803. The authors compared the spectroscopic properties of the red_a with the wild type PSI, and showed that the newly formed Chl trimer results in the red shift spectra and creates a new red site c710. They also solved the cryo-EM structure of the red_a PSI complex to show that it has highly similar structure as the wild type PSI, indicating that the formation of c710 is due to the presence of the Chl trimer B31-B32-B33. In addition, the authors

found a large-scale structural heterogeneity of the trimeric PSI, suggesting the movement of PSI monomers within a trimer.

This is a quite interesting work, and the first one to identify the red site in the PSI core, thus advances our knowledge. However, I have a major concern about the quality of the cryo-EM structure. **The model-map fit is very poor, even for the core subunits PsaA and PsaB, as shown in the PDB validation report.** It seems something wrong with the coordinate or the validation report. I require the authors to further adjust and refine their structure and validate it again when revise their manuscript. Since the authors discuss about the structural heterogeneity and the short distances between interfacial chlorophylls, the accurate structural model should be provided.

We thank reviewer #2 for his generous comments. We've improved the resolution of our map to 3.1 Å and re-refined our model into it. The new validation report as well as other measures such as Q score (see below) indicate that the map to model fit is vastly improved. We are attaching the new model and map together with the new validation report for the reviewer's examination.

Minor points - I suggest that the authors to comment whether the PSI complex from *T. elongatus* shows the same spectroscopic property of c710, since the *T. elongatus* PSI has the same Chl trimer.

T.E does have a 710 state and we have emphasized this in the revised manuscript.

- Page 11, the citation of "Supplementary figure 5" is probably wrong. I think it should be "Supplementary figure 6".

We have changed it to Supplementary figure 7 since we added additional supplementary figure.

Reviewer #3 (Remarks to the Author):

This manuscript utilized the mutagenesis approach to identify the location of a red site in the core antenna of cyanobacteria photosystem I. The authors utilized spectrometry and cryo-electron microscopy to validate their finding. I will leave other reviewers to qualify the scientific novelty and merit of this work. I only give some comments to the issue of cryoEM image processing and molecular modeling,

1. According to the validation report submitted from the authors, it seems that the current model has a very good geometry score but very poor clash score. And the reported fitness between model and EM map is rather poor with all

most residues appear poor fit. This issue should be fixed. I guess there could be at least three points to be considered. **First, the authors should think about the correctness of pixel size. Was the pixel size of the camera under the condition of data collection well calibrated using one standard specimen?**

The authors could also use some software packages to check this problem.

Second, the Ramachandran statistics and geometry of bond and angle are rather good. It seems that the authors utilized a tight geometry restriction during model refinement or performed a further round of geometry optimization after model refinement. This would induce a poor fitness of cryoEM map and is not reasonable. Third, the final cryoEM map was post-processed with a proper mask and b-factor. The authors should check whether the mask is too tight and thus some weak densities were truncated. With the above issue solved, the authors are also suggested to calculate and report the Q score of their model (chain by chain) according to the recent literature Nature methods 17, 328-334. Of course, a new validate report is needed in the next round of review.

2. The authors should provide more detailed information about cryoEM data collection. **For example, what kind of software was used to collect data automatically? Whether the beam/image shift protocol (or just stage shift) was applied during data collection? How many frames were fractionated in each exposure movie? The representative micrograph with few number of particles in S4a seems a cropped version of the original one. The authors should present the representative original one. The description of magnification and pixel size in Methods would be problematic (P22, line 18). The nominal magnification should be like 47,600X. The accuracy of pixel size should not be such high with four digits. The description of counting rate (P22, line 19) is not correct. It should be dose rate. Again, the four digits of dose rate is not rigorous.**

3. The data processing procedure should be also revised carefully with more information provided. Which version of RELION was used? The citation of RELION was missed. After Fourier-crop, the pixel size has been doubled. Thus the pixel size of 1.05 Å/Px is not yielded from particle exacting in RELION. Whether the CTF refinement (which is essential to further push resolution) was performed? Whether the procedure of particle polishing was performed? The b-factor should have a unit of Å². The authors are suggested to validate the orientation distribution effect by calculating Eod score according to the recent literature Nat Commun 8, 629.

We would like to thank reviewer number 3 for his comments on our Cryo-EM data processing, data collection and modeling. We reanalyzed our data using relion 3.1, as a result our final resolution improved to 3.1 Å. Per particle ctf refinement and polishing were performed using relion 3.1. We have included more information in our data processing diagram to be complete as possible and expanded our description of image processing, we've included additional citations regarding the per particle ctf refinement procedure in our paper. The masks used during multibody refinement are shown in supplementary figure 7a, were threshold at very low levels and included a soft edge of 8 pixels. Masks for final resolution estimation were slightly tighter (with an 0.0089 threshold) but included a 10 Å low pass filter, a 4 pixel expansion and a an 8 pixel soft edge. We show our phased randomized FSC curves to verify that these masks were not too tight and do not bias our resolution estimation.

The pixel size of the microscope was calibrated and found to be 1.05 Å. In addition, we did not notice any bumps in our FSC throughout our data processing which would indicate a large error in pixel size (~5%). To get a measure of smaller errors in pixel size we followed the procedure in the relion wiki to determine the pixel size from the estimated spherical aberration. The result was that our “real” pixel size was 1.045 Å, since our dataset's resolution is marginal for this analysis we used a higher resolution dataset (2.8 Å) taken on the same microscope (albeit a year later) and found a much smaller discrepancy in pixel size (virtually 0). We conclude that a pixel size of 1.05 Å is correct and we did not change it. Our current model shows much improved fit to the map, in addition to the pdb validation report we calculated our Q score as suggested by the reviewer and include the per chain results in supplementary table 2 in addition to the Q score of the chlorophylls and carotenes. All scores are consistent with our reported resolution. We have also analyzed our views distribution using cryoEF. Using our correct particle size, we obtained an efficiency score of 0.83 which indicates good efficiency and the output did not include any warning on missing views. We did not include this information in the manuscript as we feel this is already described well by the views distribution in supplementary figure 5b.

Our current models Clash score is still a bit high at 11.4. this probably reflects the present of the many ligands in the model. For example, the clash score of the 2.5 Å crystal structure of the *Synechocystis* sp. PCC6803 PSI is 16 (PDB: 5OY0) and there are more examples of higher clash score in PSI complexes. Our current score falls in the middle of the pack, without considering resolution. Our pdb validation report does not include a clash score due to a technical issue the with pdb validation

pipeline. We are pasting the response of the pdb curator to our query on this below:

“Hi Yuval

I have reported the clashscore issue to our developer. There was a known bug in MolProbity, it sometimes output clashes between atom pairs with distance > 5Å. To prevent such incorrectness in validation report, the validation pipeline will not produce a clashscore if the code detected a too long clash output by molprobity (> 5Å).

We are hoping the new version of MolProbity can fix the problem.

Best,

Yuhe”

In our refinement table we included the value of 11.4 for clash score which is the current output of MolProbity for our structure in both the latest phenix version and the MolProbity server.

SerialEM was used to collect the data with stage shift only. The defocuse value was varied over the indicated range, total exposure time was 8 seconds fractionated into 40 frames movies. We added this information to our data processing diagram and methods section. We have added a complete micrograph which includes more particles in supplementary figure 5b. The RELION citation is placed at the methods section ref. 12. We have added a few more citations where appropriate (for example for the per particle ctf refinement procedure).

Regarding the pixel size and binning. Our movies were taken at super resolution mode and then binned at 2X, first using MotionCor2 and then again (from the original movies) during Bayesian polishing in Relion. We have changed our super pixel size to 0.525 Å and this was derived from the calibrated pixel size of the microscope (1.05Å). Instead of stating a magnification figure we simply included the calibrated pixel size and the identity of the camera with its known physical pixel size of 5 microns in our data collection table, arriving at the correct magnification value of 47,600 is trivial and transparent in that way and can be done by any reader. We have included the Total electron dose in our table (45 e/Å²) and the dose rate in the method section with appropriate units.

Bibliography:

1. Akita, F. *et al.* Structure of a cyanobacterial photosystem I surrounded by octadecameric IsiA antenna proteins. *Commun. Biol.* **3**, (2020).
2. Gisriel, C. *et al.* The structure of Photosystem I acclimated to far-red light illuminates an ecologically important acclimation process in photosynthesis. *Sci. Adv.* **6**, eaay6415 (2020).
3. Chen, M. *et al.* Distinct structural modulation of photosystem I and lipid environment stabilizes its tetrameric assembly. *Nat. Plants* **6**, 314–320 (2020).
4. Cao, P. *et al.* Structural basis for energy and electron transfer of the photosystem I–IsiA–flavodoxin supercomplex. *Nat. Plants* **6**, 167–176 (2020).
5. Zheng, L. *et al.* Structural and functional insights into the tetrameric photosystem I from heterocyst-forming cyanobacteria. *Nat. Plants* (2019). doi:10.1038/s41477-019-0525-6

REVIEWERS' COMMENTS

Reviewer #1 (Remarks to the Author):

It's difficult to understand how two different stories (Structural Fluctuations in trimmer and red-Chl) are one paper, but the reviewer's questions were carefully answered. So, the manuscript has been much improved. This reviewer is OK with this paper being published.

Reviewer #2 (Remarks to the Author):

The authors have satisfactorily addressed my comments. The revised manuscript has been much improved, and I recommend that it will be accepted for publication.

Reviewer #3 (Remarks to the Author):

The authors have revised their manuscript fully considering my suggestions. I have no major issue regarding to the technical part.

Two minors:

1. Page 25 and Line 7, a unit of b-factor is lack and needs to be added.
2. Page 21 and Line 21, "Electro-Magnetic Database" is wrong and should be "Electron Microscopy DataBase".

REVIEWERS' COMMENTS

Reviewer #1 (Remarks to the Author):

It's difficult to understand how two different stories (Structural Fluctuations in trimmer and red-ChI) are one paper, but the reviewer's questions were carefully answered. So, the manuscript has been much improved. This reviewer is OK with this paper being published.

Reviewer #2 (Remarks to the Author):

The authors have satisfactorily addressed my comments. The revised manuscript has been much improved, and I recommend that it will be accepted for publication.

Reviewer #3 (Remarks to the Author):

The authors have revised their manuscript fully considering my suggestions. I have no major issue regarding to the technical part.

Two minors:

1. Page 25 and Line 7, a unit of b-factor is lack and needs to be added.

We have added the b-factor unit.

2. Page 21 and Line 21, "Electro-Magnetic Database" is wrong and should be "Electron Microscopy DataBase".

We corrected this sentence.